# P-BERT: Hardware-Aware Optimization of BERT Using Evolutionary Techniques

## Abstract

Transformer-based models have emerged as the go-to standards in Natural Language Processing (NLP), revolutionizing the landscape of NLP applications. As complex models continue to proliferate, the need for more efficient computational processing becomes increasingly imperative. This has led to the rise of model compression techniques, implemented to target computational inefficiencies. Expounding on this, we propose Pyramid-BERT (P-BERT), the integration of three established model compression techniques to further reduce the computational inefficiency of the standard BERT models, and subsequently optimize BERT under the hardware characteristics. Specifically, the techniques employed are pruning, quantization, and knowledge distillation. The first two aforementioned correlated techniques work simultaneously to remove redundant specifications while leveraging knowledge transfer from baseline models. These techniques enable a substantial reduction in computational cost, making P-BERT highly suitable for portable, low-power devices such as cellphones, wearable devices, and smartwatches, and thus enabling hardware-friendly processing on various computing engines. Additionally, we will be proposing a new metric, the inverted computational complexity to quantify the complexity and efficacy of the model. This metric aims to more accurately capture the hardware-specific performance characteristics. Our experimental results show that P-BERT achieves a remarkable reduction of at least 60% in the inverted computational complexity ratio while ensuring comparable accuracy and scores across many downstream tasks compared with the baseline BERT models.

## 1 Introduction

In recent years, transformer-based models emerged as the cornerstone of natural language processing (NLP), catalyzing a paradigm shift in the field. The advent of pre-trained transformer-based models sparked intense interest and research activity in both industry and academia. This led to the creation of numerous transformer-based models, including GPT-4 Achiam et al. (2023), RoBERTa Liu et al. (2019) and DistillBERT Sanh et al. (2019).

Central to the exceptional performance of these transformer-based models are their deep forward propagation in each layer. While effective, these neural networks are time and space inefficient, due to overparameterization Ba & Caruana (2014); Rogers et al. (2021) and a large number of operations involved. Additionally, given the rise diversity of resource-constrained devices, including mobile phones and edge-devices, current solutions may not be optimal Ganesh et al. (2021). Deployment of these models in such devices and majorly constrained by computation power, storage capacity and energy Gou et al. (2021).

Consequently, there are various model compression techniques proposed to mitigate computational inefficiencies of such models, including pruning, quantization, distillation, parameter sharing and module replacing Qiu et al. (2020), low-rank factorization Cheng et al. (2018), matrix decomposition Ganesh et al. (2021). Exploration of these techniques include investigating the effect of pruning and transfer learning Gordon et al. (2020) and a novel integer-only quantization approach by Kim et al. (2021). However, directly compressing these models often leads to performance loss, quantified primarily by accuracy loss.

Hence, we propose Pyramid-BERT (P-BERT), a hardware-aware optimization of the BERT model that integrates three proven model compression techniques—pruning, quantization, and knowledge distillation. By combining these techniques, P-BERT addresses the computational inefficiencies inherent in BERT while minimizing performance loss. We selected BERT as the baseline model due to its suitability for edge devices, aligning with our focus on hardware-aware optimization. Additionally, we are proposing a novel metric, the inverted computational complexity ratio, to quantify the complexity and effectiveness of model compression techniques relative to a baseline model.

Our experimental results demonstrate that P-BERT achieves at least a 60% reduction in the inverted computational complexity ratio, while minimising performance loss on downstream tasks such that the scoring metric remains comparable with the baseline BERT model. This would mean a fall in at most 5-6% accuracy or other similar metric the large majority of the time.

In summary, our contributions are highlighted as follows:

- Proposed a variant of the original BERT model, Pyramid-BERT, an integration of three established model compression techniques - pruning, quantization and knowledge distillation, that optimizes performance for hardware efficiency.

- P-BERT achieves at least a 60% reduction in inverted computational complexity ratio, well ensuring comparable accuracy and scores with the baseline model.

- A metric that quantifies the complexity and efficacy of a model — inverted computational complexity ratio.

## 2  RELATED WORKS

Many works that have delved into individual model compression techniques, such as pruning, quantization and distillation, each offering unique insights and advancements.

For instance, MiniLM proposed a deep self-attention distillation for task-agnostic Transformer based LM distillation, involving distilling the self-attention module of the last Transformer layer of the teacher model Wang et al. (2020). Wang et al. conducted further research with MiniLMv2, introducing multihead self-attention relation distillation, which allows fine-grained self-attention knowledge and thus flexibility in the number of student's attention heads Wang et al. (2021a). Patient Knowledge Distillation proposed an approach that capitalizes on patient learning from multiple intermediate layers of the teacher model Sun et al. (2019). The approach employs two unique strategies — learning from the last $k$ layers and learning from every $k$ layer. Their approach improved results on multiple NLP tasks, with a significant reduction in the total parameters and inference times. However, using the following metrics as a measure of computational efficiency has its limitations that have been discussed widely in literature reviews Cheng et al. (2018).

Quantized Neural Networks trains networks with low precision weights (1-bit) and activations Hubara et al. (2018). They reduced memory size, and accesses and replaced most arithmetic operations with bit-wise operations, while achieving comparable accuracy to the baseline model. Regarding quantization, Shen et al. proposed a method for quantizing BERT models to ultra-low precision through a new group-wise quantization scheme and a Hessian-based mix-precision method Shen et al. (2020). Quantized 8Bit BERT (Q8BERT) utilizes quantization-aware training during BERT's fine-tuning phase to compress the model by a factor of 4 with negligible accuracy degradation, while also enhancing inference speed on hardware that supports 8-bit integer operations Zafrir et al. (2019). Other state-of-the-art quantization compression methods include GPTQ Frantar et al. (2022), AWQ Lin et al. (2024), AQLM Egiazarian et al. (2024), and QuIP# Tseng et al. (2024) leveraged advanced techniques to achieve significant reductions in model size while maintaining performance across various natural language processing tasks.

Similarly, some works integrated two model compression techniques. Polino et al. proposed two new methods: quantized distillation and differentiable quantization, both jointly leverage weight quantization and knowledge distillation Polino et al. (2018). The first incorporates distillation loss into quantized weights, while the second optimizes the location of quantization points using stochastic gradient descent. Evolutionary Multi-Objective Model Compression proposed combining pruning and quantization to optimize energy efficiency and accuracy simultaneously Wang et al. (2021b).

Through integrating pruning, quantization, and knowledge distillation, our approach balances efficiency and effectiveness, while proposing a novel metric to evaluate the computational efficiency and complexity of models.

## 3 BACKGROUND

### 3.1 PRUNING

Deep neural networks are recognized to be overparameterized Ba & Caruana (2014); Rogers et al. (2021), leading to the high computational and memory cost. Pruning involves the intentional reduction of redundant parameters Cheng et al. (2018); Ganesh et al. (2021). Pruning is often explored in relation to the lottery ticket hypothesis in neural networks Frankle & Carbin (2018) and has been discussed in the context of BERT Chen et al. (2020).

### 3.2 QUANTIZATION

The high computational inefficiency of deep neural networks is due to the large amount of multiply-accumulate operations involved in the computation. Quantization directly targets this by reducing the bit-size of parameter values, which leads to requiring less computational power, memory storage, and inference time Kim et al. (2021).

### 3.3 KNOWLEDGE DISTILLATION

Knowledge Distillation, specifically Task-Specific Distillation, is the third pillar in our proposed model. The method of knowledge distillation distills knowledge from the larger teacher model into the smaller student model Hinton et al. (2015). Knowledge distillation varies across knowledge categories, training schemes, teacher-student architecture, and distillation algorithms Gou et al. (2021). Specific to P-BERT, we adopted a layer-wise knowledge distillation technique by Neo et al. Ming et al. (2023), to have the student model replicate the teacher model's behavior. The details of this approach are elaborated below.

#### 3.3.1 OBJECTIVE FUNCTION

The cross-entropy (CE) between the student's logits $\mathbf{z}^s$ and true labels $\mathbf{y}$ penalizes wrong classifications is given in Equation (1).

$$L_{\text{hard}} = CE(\mathbf{z}^s, \mathbf{y}) \tag{1}$$

A higher hyperparameter temperature, $t$, leads to a smoother probability distribution. The cross entropy between student's logits and teacher's logits, $\mathbf{z}^T$ is given in Equation (2).

$$L_{\text{soft}} = CE(\mathbf{z}^s/t, \mathbf{z}^T/t) \tag{2}$$

Equation (3), integrated the mean-squared error (MSE) to ensure each layer of the student learns from the corresponding layer of the teacher. $\mathbf{H}_i^S$ and $\mathbf{H}_i^T$ indicates the hidden states of the student and teacher models respectively, where $l$ is the number of layers in the model.

$$L_{\text{hidn}} = \sum_{i=0}^{l} MSE(\mathbf{H}_i^S, \mathbf{H}_i^T) \tag{3}$$

The discrepancies between the student's attention output $\mathbf{A}_i^S$ and $\mathbf{A}_i^T$ is penalised by Equation (4).

$$L_{\text{attn}} = \sum_{i=1}^{l} MSE(\mathbf{A}_i^S, \mathbf{A}_i^T) \tag{4}$$

The combined overall objective function for knowledge transfer is summarized in Equation (5), where $\alpha$, $\beta$ and $\gamma$ are hyperparameters to be tuned.

$$L_{\text{net}} = (1 - \alpha) * L_{\text{hard}} + \alpha * L_{\text{soft}} + \beta(L_{\text{hidn}} + \gamma * L_{\text{attn}}) \tag{5}$$

# 4 METHODOLOGY

## 4.1 UNSTRUCTURED PRUNING

This approach greedily prunes hidden states in the feed-forward neural network layers of the transformer architecture that are deemed less meaningful. The following method determines which hidden states to remove. It is illustrated in Fig (1) as a flowchart. An example of it is in Fig (2).

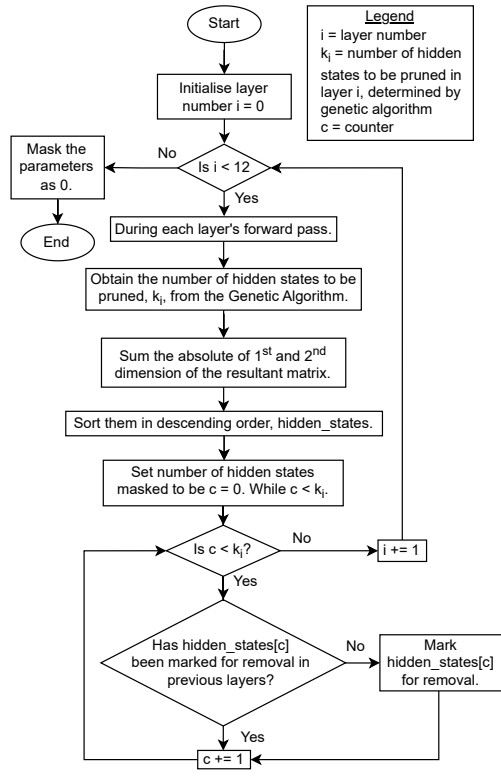

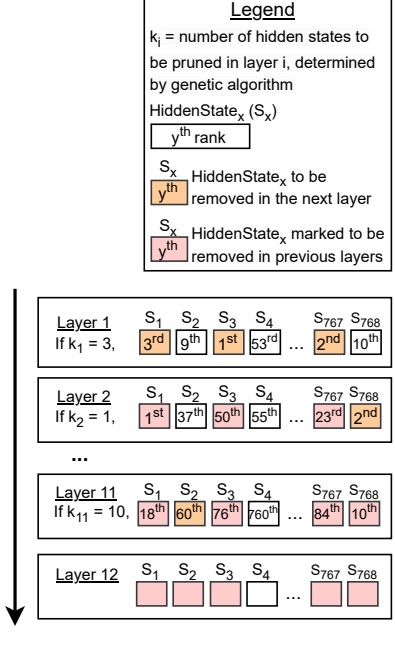

Figure 2: An example of 4.1 Pruning. Layer 1 has $k_1 = 3$ states to remove and hidden states $S_3$, $S_{767}$, $S_1$ are the 3 with the lowest absolute sum, and thus, marked for pruning (orange). Similarly, layer 2, with $k_2 = 1$, $S_{768}$ is marked for pruning despite being the second highest as $S_1$, the highest, was marked to be pruned in a previous layer (red). This ends in layer 11. In layer 12, those in red are subsequently masked.

Figure 1: Flowchart of 4.1 Pruning. $k_i$, the number of hidden states is initialised for each layer, i, by the genetic algorithm.

1. The number of hidden states to be pruned at each layer $i$ , $k_i$, is initialized through genetic algorithm, where $1 \leq k_i \leq 69$.

2. For every layer, where $1 \leq i \leq 11$, the $k_i$ hidden states with the lowest absolute summation of the $1^{st}$ and $2^{nd}$ dimensions of the weight matrix are greedily marked for pruning. Hidden states previously marked for removal cannot be marked again, thus the next unmarked states will be selected again.

3. The pruning of the hidden states is simulated through the masking of both the pre-trained weight matrices of the baseline BERT model and subsequently computed matrices. The pruning begins with the $2^{nd}$ layer's feed-forward.

## 4.2 QUANTIZATION

BERT follows a standard 23-bit. In our proposed method, each layer can be quantized to: 4-bit, 8-bit, 16-bit, or 32-bit. Quantizing at various levels reduces the model size and computation requirements. Quantizing to lower bit widths, such as 4-bit or 8-bit, leads to compression by reducing the number

of bits needed to represent the model's parameters. The remaining bits in the 32-bit scenario are used for the scale. The number of bits per layer for each model is initialized through genetic algorithm. Our implementation of quantization is performed simultaneously with the pruning, and includes leveraging a straight-through estimator to allow back-propagation during training.

### 4.3 INVERTED COMPUTATIONAL COMPLEXITY RATIO

We propose a new metric for evaluating the computational efficiency and complexity of a transformer-based model — inverted computational complexity ratio. The computational complexity is computed as follows: for each layer, beginning with the second, we obtain the product of the layer number, $i$, the total number of hidden states left in that layer after pruning, $j_i$, and the number of bits in that layer, $b_i$. The product for each layer is then summed for the remaining layers (11 in the case of BERT) to give the computational complexity metric. Equation (6) illustrates the computation of the computational complexity, $K_{\text{model}}$.

$$K_{\text{model}} = \sum_i ij_ib_i \tag{6}$$

#### 4.3.1 COMPUTATIONAL COMPLEXITY OF BASELINE BERT

For example, BERT has $b_i = 23$, $j_i = 768$ for all $i \subset \{1, 2, ..., 11\}$. It has 11 layers excluding the first, each layer with a 23-bit size, and a constant of 768 hidden states in each layer. We can obtain the computational complexity of BERT, $K_{\text{BERT}}$, using Equation (7).

$$K_{\text{BERT}} = \sum_i ij_ib_i = 1165824 \tag{7}$$

Using the computational complexity of the baseline as a reference, point we can compute the inverted computational complexity ratio as follows in Equation (8).

$$\eta = \frac{K_{\text{BERT}}}{K_{\text{model}}} \tag{8}$$

Figure 3 and Figure 4 both depict a downward sloping straight line obtained using linear regression

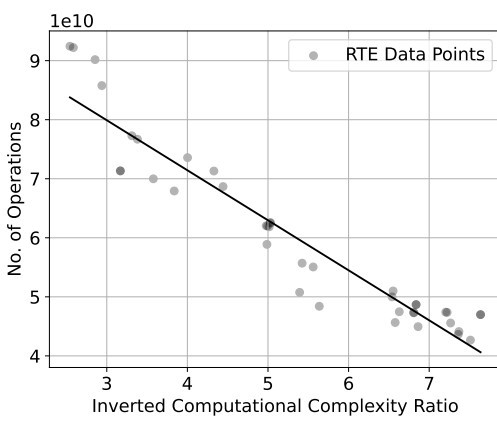
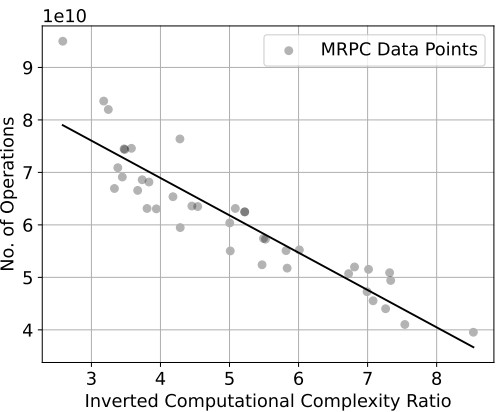

Figure 3: Scatterplot of No. of Operations against Inverted Computational Complexity Ratio using results from GLUE Task RTE's final population of the genetic algorithm in 5. Experiments.

Figure 4: Scatterplot of No. of Operations against Inverted Computational Complexity Ratio using results from GLUE Task MRPC's final population of the genetic algorithm in 5. Experiments.

that minimizes squared error. This illustrates a linear correlation between the inverted computational complexity and the number of operations. As the inverted computational complexity ratio increases, implying a less complex proposed model, the number of operations decreases, similarly implying that the model requires fewer operations. In both cases, a model that has fewer hidden states or layers than the original BERT model, would then require comparably fewer operations and thus be a less complex model. Hence, both metrics are inversely proportional.

### 4.3.2 Discussion of Inverted Computational Complexity Ratio

While the FLOPs metric is a widely used indicator of computational complexity, it does not provide a complete picture of energy efficiency or consumption, particularly in the context of quantization. FLOPs measure the number of floating-point operations required, but they fail to account for the impact of model optimizations, such as quantization, that use lower precision arithmetic like 8-bit integers. Quantization can significantly reduce both memory usage and energy consumption, but these benefits are not reflected in the FLOPs measurement. Consequently, while our proposed *Inverted Computational Complexity Ratio* offers valuable insight into model compression, it also highlights the limitations of FLOPs as an indicator of real-world efficiency, particularly in hardware-accelerated environments where lower precision operations dominate.

### 4.4 Genetic Algorithm

We chose to leverage the genetic algorithm, specifically Nondominated Sorting Genetic Algorithm II Deb et al. (2002), to select optimal models. Our objective for the genetic algorithm first prioritizes the evaluation metric obtained, followed by the inverted computational complexity. The genetic algorithm is utilized to determine the number of hidden states that are to be pruned at each stage of the BERT model, and the bit size that each layer would be using. The values are initialized randomly.

Following the computation of the fitness score of each individual in the population, the new population is selected from individuals with higher fitness scores and obtained through crossover and mutation. A detailed flowchart integrating pruning and quantization at each layer is illustrated in Fig (5).

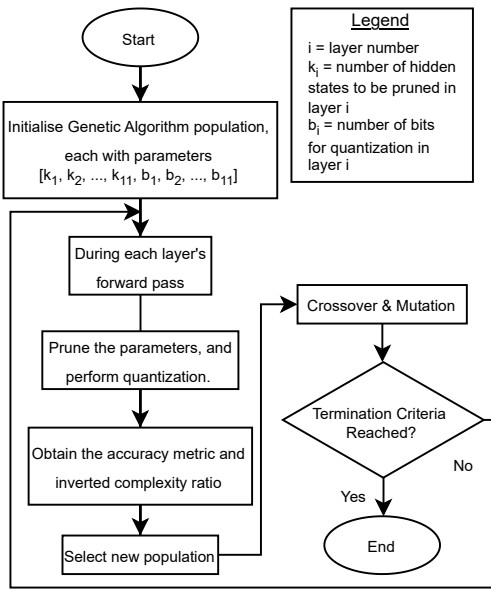

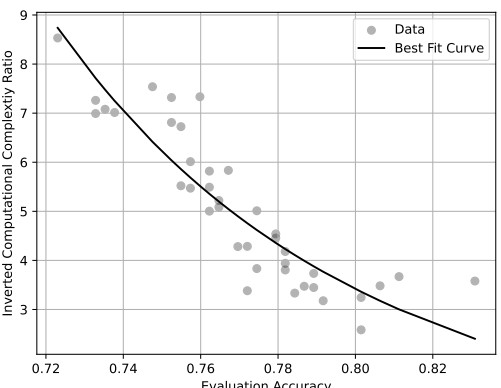

Figure 6: Pareto Front of the Final Population from the Genetic Algorithm on GLUE Task MRPC. Knowledge distillation has yet to be applied here. A power law curve was found to fit the scatter plot of evaluation accuracy and inverted computational complexity ratio the best. As seen in the figure above, the data points are scattered evenly around the fitted curve. Additionally, there is an observable downward trend, where the evaluation accuracy is seen to increase with a decrease in the inverted computational complexity ratio. This would suggest that the more complex the model is, the higher the evaluation accuracy, which falls in line with existing literature.

Figure 5: Flowchart of the Genetic Algorithm with the integration of the pruning and quantization steps. This illustrates our utilization of the genetic algorithm to determine the number of hidden states to be pruned and the bit size of each layer. Subsequently, the genetic algorithm would select individuals with a maximized accuracy metric and inverted complexity ratio for crossover and mutation.

### 4.5 Knowledge Distillation: Task-Specific

The simultaneous pruning and quantization introduced a performance gap between the proposed model and the baseline model. To bridge this gap, we leveraged task-specific knowledge distillation.

- **Teacher Model:** BERT baseline model.
- **Student Model:** The proposed pruned and qunatized model.

Once the optimal models have been selected using the genetic algorithm, we employed knowledge distillation with tuned hyperparameters to obtain the final score.

## 5 Experiments

We evaluated our approach on Hugging Face's text classification datasets, specifically the GLUE benchmarks tasks RTE, MRPC, STSB, and CoLA. The evaluation metrics we have chosen are accuracy for RTE and MRPC, Pearson Coefficient for STSB, and Matthews Correlation Coefficient for CoLA. Our experiments are conducted on AMD EPYC Millan 7713 for CPU and NVIDIA A100-40G SXM for GPU.

For our genetic algorithm, we set the population size = 40 and the number of generations = 5. For our GLUE tasks, the input sequence length = 128 tokens, number of epochs = 4, training batch size = 32, and a learning rate of 5e-5.

### 5.1 Pruning and Quantization Results

We chose GLUE Task MRPC as the representative pareto front in Fig (6) because MRPC's dataset contains a significant number of examples more than the GLUE Task RTE. Its evaluation metric of accuracy is preferred over STSB's Pearson Correlation and CoLA's Matthews Correlation Coefficient for its straightforward assessment of the model's classification capabilities.

Tables 1, 2, 3 and 4 illustrate the optimal final population from the genetic algorithm for our chosen GLUE tasks RTE, MRPC, STSB and CoLA. It compares combinations of removed hidden states and quantized layers against each other and the baseline BERT model. As observed from all tables, as the inverted computational complexity ratio increases the respective metric columns for the four tasks (Acc., Acc., $\rho$, and MCC) similarly decrease. From these tables, we can also observe the effectiveness of pruning and quantization. This is indicated by the high inverted computational complexity ratio and low number of operations and low estimated inference time.

Notably, while a general decreasing trend in the number of operations (normalized) is observed, variability persists in tables 1, 2, 3 and 4. The number of operations does not fully capture the effects of quantization. A model exhibiting both a higher inverted computational complexity ratio and a higher number of operations would indicate a model that is quantized to a greater extent and pruned to a lesser extent. This indicates that quantization, while reducing computational complexity, is not accurately reflected in operation counts, highlighting the need for more comprehensive metrics when evaluating model efficiency.

The Estimated Inference Time (s) column in the tables reflects the normalized number of operations and is derived based on our GPU's performance, which operates at 19.5 TFLOPS for FP32 precision. This estimation covers the entire dataset, including training, validation, and test examples for each task. As observed in tables 1, 2, 3 and 4, the inference times show variation despite the decreasing computational complexity ratio. This discrepancy arises because FLOPs primarily measure floating-point operations, whereas our model predominantly utilizes integer operations. Consequently, the estimated inference time has limitations, as it does not account for the impact of quantization.

### 5.2 Knowledge Distillation Loss Curve

To construct the knowledge distillation loss curve, we modified the number of epochs = 10, while keeping the rest. This is because we can better observe the loss curve with more epochs. We have standardized our hyperparameters for the knowledge distillation loss curve: $\alpha = 0.9$, $\beta = 1.0$, temperature, $T = 15.0$. This imitates the hyperparameter setup in MA-BERT Ming et al. (2023), as we

replicated their distillation technique in our methodology. We have chosen to fix the hyperparameters to ensure that the scale of the loss remains comparable within each task.

### 5.2.1 RESULTS

Each combination of the results from the optimal final population of the genetic algorithm for RTE and MRPC (as shown in Table 1 and 2) are indicated by the low opacity black curves in Fig (7) and Fig (8) below. The solid black curves depict the mean of all curve losses for that specific task. The mean loss curve is then smoothed under a smooth moving average and depicted in orange.

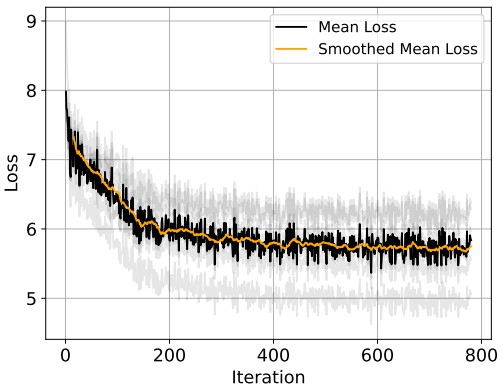 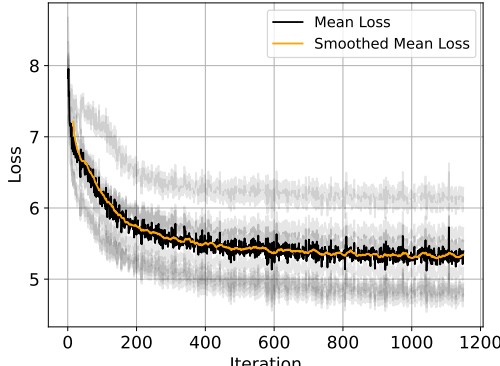

Figure 7: Graph of knowledge distillation loss against the number of iterations on GLUE Task RTE's final optimal population. A gradual decrease in the mean loss and smoothed mean loss is observed across the first 200 iterations before stagnating over the remaining 600 iterations.

Figure 8: Graph of knowledge distillation loss against the number of iterations on GLUE Task MRPC's final optimal population. A gradual decrease in the mean loss and smoothed mean loss is observed across the first 400 iterations before stagnating over the remaining 800 iterations.

### 5.3 KNOWLEDGE DISTILLATION & HYPERPARAMETER TUNING

Our hyperparameters are tuned using Python package optuna's GridSampler, a variation of Grid Search. The search space of $\alpha = (0.0, 1.0)$, $\beta = [0.0, 10.0]$, $T = [0.0, 20.0]$.

### 5.3.1 RESULTS

Our knowledge distillation results are performed on the final population of each GLUE task. Each member of the final optimal population has a different underlying architecture with different hidden states pruned and bits quantized. Hence, each model in the optimal final population underwent hyperparameter tuning with knowledge distillation, leading to different hyperparameters. Details on which hyperparameter was selected for each model can be found in A.1.

Tables 1, 2, 3 and 4 largely depict an increase in their evaluation metric following knowledge distillation with tuned hyperparameters, which is the 5th column. Across all tables, models with higher inverted computational complexity ratios had lower scores on their respective evaluation metrics. However, after knowledge distillation and hyperparameter tuning, these values all rose, with the highest $\eta$ of each task increasing the most. This is most significant in table 4 with a 13.6 increase.

However, tables 1 and 2 where the top 2 and 3 accuracies in each task respectively show a fall or stagnating accuracy instead. These models have the lowest inverted computational complexity ratios in their task, indicating a more complex model. We believe that this is due to the limited room for improvement in the score metric that knowledge distillation can bring about. It is possible that this is the result of an insufficient number of hyperparameters, or the lack of precision in the hyperparameters (i.e., the number of decimal places). Additionally, the accuracy we have obtained after knowledge distillation for RTE remains within 5.5% of the original baseline model, even while using a significantly less complex model.

Table 1: RTE Optimal Final Population

| $\eta^*$ | No. of Operations (Norm.) | Est. Inference Time (s) | Acc. | Acc. After KD |
|---|---|---|---|---|
| **3.38** | 0.678 | 17.2 | 66.8 | 66.4 |
| 5.02 | 0.704 | 17.9 | 65.7 | 65.7 |
| 7.21 | 0.553 | 14.0 | 65.0 | 65.3 |
| 7.27 | 0.589 | 14.9 | 62.8 | 64.6 |
| 7.37 | 0.606 | 15.4 | 62.1 | 65.0 |
| 7.64 | 0.475 | 12.1 | 59.9 | 65.3 |
| 1** | 1 | 25.4 | 70.8 | |

*$\eta$ represents Inverted Computational Complexity Ratio
**BERT has $\eta = 1$ computed from Equation 8

Table 2: MRPC Optimal Final Population

| $\eta^*$ | No. of Operations (Norm.) | Est. Inference Time (s) | Acc. | Acc. After KD |
|---|---|---|---|---|
| **3.58** | 0.847 | 21.6 | 83.1 | 80.6 |
| 3.67 | 0.755 | 19.3 | 81.1 | 79.4 |
| 3.73 | 0.682 | 17.4 | 78.9 | 78.2 |
| 4.18 | 0.622 | 15.9 | 78.2 | 79.4 |
| 4.54 | 0.709 | 18.1 | 77.9 | 79.2 |
| 5.01 | 0.754 | 19.2 | 77.5 | 79.7 |
| 5.83 | 0.669 | 17.1 | 76.7 | 77.9 |
| 7.33 | 0.559 | 14.3 | 76.0 | 77.0 |
| 7.54 | 0.589 | 15.0 | 74.8 | 76.2 |
| 8.53 | 0.586 | 15.0 | 72.3 | 75.7 |
| 1** | 1 | 25.5 | 86.0 | |

Table 3: STSB Optimal Final Population

| $\eta^*$ | No. of Operations (Norm.) | Est. Inference Time (s) | $\rho^{***}$ | $\rho^{***}$ After KD |
|---|---|---|---|---|
| **2.47** | 0.633 | 24.0 | 87.2 | 87.2 |
| 3.57 | 0.690 | 26.2 | 86.9 | 86.9 |
| 3.96 | 0.771 | 29.3 | 86.9 | 86.9 |
| 5.05 | 0.682 | 25.9 | 86.8 | 86.3 |
| 5.15 | 0.645 | 24.5 | 86.5 | 86.5 |
| 5.25 | 0.661 | 25.1 | 86.2 | 86.3 |
| 5.51 | 0.612 | 23.2 | 86.0 | 85.9 |
| 6.97 | 0.651 | 24.7 | 85.9 | 86.1 |
| 8.49 | 0.577 | 21.9 | 84.7 | 85.6 |
| 8.67 | 0.637 | 24.2 | 83.7 | 84.2 |
| 1** | 1 | 38.0 | 89.3 | |

***$\rho$ represents Pearson Correlation

Table 4: CoLA Optimal Final Population

| $\eta^*$ | No. of Operations (Norm.) | Est. Inference Time (s) | MCC | MCC After KD |
|---|---|---|---|---|
| **3.79** | 0.828 | 38.8 | 52.6 | 57.3 |
| 4.08 | 0.814 | 38.1 | 51.1 | 55.5 |
| 4.12 | 0.784 | 36.7 | 47.9 | 55.2 |
| 4.22 | 0.822 | 38.5 | 45.6 | 54.3 |
| 4.39 | 0.678 | 31.8 | 44.4 | 48.0 |
| 5.00 | 0.677 | 31.7 | 43.5 | 45.4 |
| 5.17 | 0.776 | 36.4 | 43.0 | 50.3 |
| 5.18 | 0.741 | 34.7 | 40.1 | 49.9 |
| 5.23 | 0.632 | 29.6 | 38.0 | 46.3 |
| 5.35 | 0.761 | 35.7 | 35.5 | 48.1 |
| 5.89 | 0.693 | 32.5 | 34.0 | 44.7 |
| 6.42 | 0.677 | 31.7 | 29.2 | 42.8 |
| 1** | 1 | 46.9 | 58.4 | |

****MCC represents Matthews Correlation

In table 3, the increase in the Pearson Correlation after knowledge distillation is small. We believe that this is because there is limited room for improvement in pearson correlation, especially so given that the scores are all within a 5.1% drop when compared with the baseline BERT model. Noticeably, hyperparameters $\alpha$ and $\beta$ are largely on the same scale indicating that hard and soft loss, and hidden states and attention loss are significant on the same scale.

## 5.4 COMPARING AGAINST OTHER MODELS

In this section, we are comparing P-BERT against other BERT-variant models, including BERT Devlin (2018), RoBERTa Liu et al. (2019), DistilBERT Sanh et al. (2019), ALBERT Lan (2019), TinyBERT Jiao et al. (2019), I-BERT Kim et al. (2021) and MiniLM Wang et al. (2020). We have computed the inverted computational complexity ratio for all models using Equation 6 and Equation 8, and computed the estimated inference time(s) on the same scale as before.

Our study aimed to evaluate P-BERT against other state-of-the-art compression methods, including LayerDrop Fan et al. (2019), GPTQ Frantar et al. (2022), AWQ Lin et al. (2024), AQLM Egiazarian et al. (2024), QuIP# Tseng et al. (2024). However, we encountered challenges in implementating the following methods for BERT-based models. This unfortunately hampered our ability to perform comprehensive comparative evaluations.

The results presented in Table 5 illustrate the performance and estimated inference time of the aforementioned models on the RTE, MRPC, STSB, and CoLA datasets. Our proposed P-BERT shows promising results with competitive accuracy, particularly in CoLA, where it achieves a score of 57.3. P-BERT demonstrates a balance between accuracy and efficiency, with inference times rang-

ing from 17.2 seconds to 38.8 seconds, depending on the task, as well as a generally competitive inverted computational complexity ratio. These results underscore the effectiveness of our integrated model compression techniques in maintaining performance while optimizing computational resources, highlighting the potential of P-BERT as a viable alternative for resource-constrained environments.

Table 5: Results of Other Models

| Model | $\eta$ | Results | | | | Estimated Inference Time (s) | | | |
|---|---|---|---|---|---|---|---|---|---|
| | | RTE | MRPC | STSB | CoLA | RTE | MRPC | STSB | CoLA |
| BERT$_{base}$ | 1.0 | 70.8 | 86.0 | 89.3 | 58.4 | 25.4 | 25.5 | 38.0 | 46.9 |
| RoBERTa$_{base}$ | 1.0 | 72.6 | 89.5 | 90.7 | 57.0 | 25.4 | 25.5 | 38.0 | 46.9 |
| DistilBERT | 4.4 | 62.5 | 82.8 | 87.5 | 53.8 | 12.7 | 12.8 | 19.0 | 23.4 |
| ALBERT$_{base}$ | 1.0 | 69.7 | 88.0 | 90.7 | 53.2 | 22.7 | 22.8 | 33.9 | 41.9 |
| TinyBERT$_4$ | 27.1 | 67.2 | 85.5 | 87.4 | 17.0 | 1.8 | 1.8 | 2.6 | 3.3 |
| I-BERT | 2.9 | 65.0 | 88.7 | 90.9 | 60.0 | 0.8 | 0.8 | 1.2 | 1.5 |
| MiniLM$_{12}$ | 2.0 | 69.7 | 87.3 | 87.7 | 55.6 | 7.6 | 7.6 | 11.4 | 14.0 |
| **P-BERT** | 3.38 | 66.4 | - | - | - | 17.2 | - | - | - |
| **P-BERT** | 3.58 | - | 80.6 | - | - | - | 21.6 | - | - |
| **P-BERT** | 2.47 | - | - | 87.2 | - | - | - | 24.0 | - |
| **P-BERT** | 3.79 | - | - | - | 57.3 | - | - | - | 38.8 |

## 5.5 LIMITATIONS

One limitation our model faces is its sensitivity to initialization. Since the seeding of the genetic algorithm is randomized, it can result in variations in performance across different runs.

Another limitation is that our model is optimized specifically for lightweight, low-power devices. As a result, the performance gains and efficiencies we demonstrate may not scale effectively to larger, more powerful hardware platforms, limiting its broader applicability to high-end systems, such as GPT.

## 6 CONCLUSION

In this paper, we introduced P-BERT, a hardware-aware and computationally efficient version of BERT that integrates pruning and quantization to reduce computational complexity and knowledge distillation to maintain comparable performance to baseline BERT. We introduced a new metric — inverted computational complexity ratio, that compares the computational complexity of a model to a baseline. Our experimental results show that our approach can reduce between 60% to 88% in the inverted computational complexity ratio.

For future work, it would be valuable to explore the applicability of P-BERT on different hardware architectures and across a wider variety of datasets and benchmarks to further validate its hardware efficiency and generalizability.

## 7 REPRODUCIBILITY STATEMENT

All datasets and pre-trained checkpoints used in this study are publicly available and can be easily downloaded, facilitating straightforward reproduction of the results. The source code for this work is provided in the accompanying zip file. It includes a detailed README.md outlining the necessary steps to replicate our experiments.

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

# A APPENDIX

## A.1 HYPERPARAMETER TUNING RESULTS

Table 6: Comparing KD Hyperparameter Tuning Results for RTE With Other Models

| $\eta^*$ | Acc. | Acc. After KD | $\alpha$ | $\beta$ | $T$ |
|---|---|---|---|---|---|
| 3.38 | 66.8 | 66.4 | 0.1 | 0.1 | 8.0 |
| 5.02 | 65.7 | 65.7 | 0.1 | 0.5 | 15.0 |
| 7.21 | 65.0 | 65.3 | 0.1 | 0.1 | 5.0 |
| 7.27 | 62.8 | 64.6 | 0.1 | 0.1 | 9.0 |
| 7.37 | 62.1 | 65.0 | 0.5 | 0.1 | 10.0 |
| 7.64 | 59.9 | 65.3 | 0.2 | 1.0 | 5.0 |
| (BERT) 1[**] | | 70.8 | | | |

[*]$\eta$ represents Inverted Computational
   Complexity Ratio
[**]BERT has $\eta = 1$ computed from Equation 8
[***]PKD represents PatientKD$_6$ Sun et al. (2019). It has

Table 7: Comparing KD Hyperparameter Tuning Results for MRPC With Other Models

| $\eta^*$ | Acc. | Acc. After KD | $\alpha$ | $\beta$ | $T$ |
|---|---|---|---|---|---|
| 3.58 | 83.1 | 80.6 | 0.5 | 0.1 | 20.0 |
| 3.67 | 81.1 | 79.4 | 0.1 | 8.0 | 9.0 |
| 3.73 | 78.9 | 78.2 | 0.1 | 10.0 | 8.0 |
| 4.18 | 78.2 | 79.4 | 0.5 | 3.0 | 8.0 |
| 4.54 | 77.9 | 79.2 | 0.5 | 3.0 | 1.0 |
| 5.01 | 77.5 | 79.7 | 0.5 | 3.0 | 1.0 |
| 5.83 | 76.7 | 77.9 | 0.1 | 0.5 | 1.0 |
| 7.33 | 76.0 | 77.0 | 0.5 | 3.0 | 1.0 |
| 7.54 | 74.8 | 76.2 | 0.5 | 3.0 | 12.0 |
| 8.53 | 72.3 | 75.7 | 0.5 | 0.5 | 1.0 |
| (BERT) 1[**] | | 86.0 | | | |

Table 8: Comparing KD Hyperparameter Tuning Results for STSB With Other Models

| $\eta^*$ | $\rho^{**}$ | $\rho^{**}$ After KD | $\alpha$ | $\beta$ | $T$ |
|---|---|---|---|---|---|
| 2.47 | 87.2 | 87.2 | 0.02 | 0.01 | 0.1 |
| 3.57 | 86.9 | 86.9 | 0.02 | 0.03 | 0.1 |
| 3.96 | 86.9 | 86.9 | 0.03 | 0.01 | 0.1 |
| 5.05 | 86.8 | 86.3 | 0.01 | 0.09 | 0.1 |
| 5.15 | 86.5 | 86.5 | 0.03 | 0.01 | 0.1 |
| 5.25 | 86.2 | 86.3 | 0.03 | 0.01 | 0.1 |
| 5.51 | 86.0 | 85.9 | 0.03 | 0.03 | 0.1 |
| 6.97 | 85.9 | 86.1 | 0.1 | 0.01 | 0.1 |
| 8.49 | 84.7 | 85.6 | 0.03 | 0.03 | 0.1 |
| 8.67 | 83.7 | 84.2 | 0.1 | 0.3 | 0.1 |
| (BERT) 1[***] | | 89.3 | | | |

[*]$\eta$ represents Inverted Computational
   Complexity Ratio
[**]$\rho$ represents Pearson Correlation
[***]BERT has $\eta = 1$ computed from Equation 8

Table 9: Comparing KD Hyperparameter Tuning Results for CoLA With Other Models

| $\eta^*$ | MCC | MCC After KD | $\alpha$ | $\beta$ | $T$ |
|---|---|---|---|---|---|
| 3.79 | 52.6 | 57.3 | 0.6 | 1.0 | 10.0 |
| 4.08 | 51.1 | 55.5 | 0.4 | 1.0 | 20.0 |
| 4.12 | 47.9 | 55.2 | 0.6 | 1.0 | 5.0 |
| 4.22 | 45.6 | 54.3 | 0.5 | 1.0 | 5.0 |
| 4.39 | 44.4 | 48.0 | 0.5 | 1.0 | 1.0 |
| 5.00 | 43.5 | 45.4 | 0.5 | 1.0 | 15.0 |
| 5.17 | 43.0 | 50.3 | 0.5 | 1.0 | 20.0 |
| 5.18 | 40.1 | 49.9 | 0.6 | 1.0 | 10.0 |
| 5.23 | 38.0 | 46.3 | 0.5 | 1.0 | 1.0 |
| 5.35 | 35.5 | 48.1 | 0.1 | 10.0 | 20.0 |
| 5.89 | 34.0 | 44.7 | 0.1 | 10.0 | 20.0 |
| 6.42 | 29.2 | 42.8 | 0.5 | 1.0 | 1.0 |
| (BERT) 1[**] | | 58.4 | | | |

[****]MCC represents Matthews Correlation
   Coefficient

