# OpenReview forum: "P-BERT: Hardware-Aware Optimization of BERT Using Evolutionary Techniques"
_ICLR.cc/2025/Conference — ICLR 2025 Conference Withdrawn Submission_

### Official Review · Reviewer_VtUT · 2024-10-22

**Soundness:** 3
**Presentation:** 3
**Contribution:** 1
**Rating:** 3
**Confidence:** 1

**Summary:**

In this paper, authors proposed a method to combine multiple techniques to reduce the original Bert model to an efficient model for computational restricted hardwares and remain accuracy. The author also proposed a metric "Inverted Computational Complexity Ratio" to measure the efficiency of compressed Bert models. The methods shows reasonable results in accuracy and efficiency compared to baseline models, but it doesn't out perform all the baseline models.

**Strengths:**

- Explored the method of combing three techniques to reduce the Bert model to an efficient model.
- Proposed a metric "Inverted Computational Complexity Ratio", by calculating the ratio of weighted summation of quantization bits and remaining number of values in each layer.
- Showed complete experiments and provided ablation study in details for different ratios on multiple tasks.

**Weaknesses:**

Weaknesses:

- Not Sufficient Discussion of Original Innovation:
The proposed method is not showing enough discussion of original innovation, as it is essentially a combination of existing methods—pruning, quantization, and knowledge distillation. It would be helpful if authors could clarify the improvements of their proposed method compared to previous approaches.
	- The use of unstructured pruning with a genetic algorithm is a well-established technique for model pruning and has been extensively studied in prior works across multiple models. For example, see Multi-Objective Pruning for CNNs Using Genetic Algorithm (Chuanguang, et al., https://arxiv.org/pdf/1906.00399), and Pruning Decision Tree Using Genetic Algorithms(Jie Chen, et al., https://ieeexplore.ieee.org/document/5376632)
	- Similarly, quantization and knowledge distillation are applied in a straightforward manner. These techniques have already been proposed and thoroughly investigated for transformer-based models in earlier works, as referenced by the authors in the related works section.

- Missing Design Reasoning of the Inverted Computational Complexity Ratio:
The proposed inverted computational complexity ratio could benefit from clearer mathematical reasoning and further justification.
	- The ratio is based on K_model, defined as the summation of i \times j_i \times b_i, where i represents the layer index. This makes the metric a weighted sum where the layer index serves as the weight. The issue with this definition is that K_model becomes disproportionately sensitive to deeper layers. For instance, Layer 11 contributes more to this metric when it is pruned or quantized, compared to other layers. It would be valuable for the authors to offer more insight and intuition into why deeper layers are weighted more heavily in their metric, and how this aligns with real-world computational overall efficiency improvements.

- Performance is Not Significant Compared to Other Models:
The performance conclusion claimed by the authors is not sufficient enough when compared to other models.
	- The authors stated that their method achieved “promising results with competitive accuracy, particularly in CoLA”, but the evidence presented does not sufficiently demonstrate accuracy or efficiency advantages over competing models on multiple tasks, particularly when compared to models like TinyBERT and I-BERT. For instance, in Table 5, TinyBERT, despite having a much larger inverted computational complexity ratio, 27.1, outperforms P-BERT on accuracy in all tasks except CoLA. Moreover, I-BERT, which has a inverted computational complexity ratio of 2.9, similar level to P-Bert, still outperforms P-BERT in tasks such as MRPC, STSB, and even CoLA. It would be great to add more discussion of the performance gap when comparing to other baselines. Discussing both the advantages and disadvantages will offer a more balanced view and help readers better understand the specific contributions and limitations of the proposed approach.
    - The authors stated that the method is “hardware-aware optimization”, but the paper does not discuss much about the meaning of "hardware-aware" and how their approach is optimized for "hardware-aware". Assuming the authors are referring to “low-computational resource hardware,” the paper does not discuss its advantages and disadvantages compared to other baseline methods optimized for hardware. For example, one of the baseline models, I-BERT, which uses “integer-only distillation” shows advantages when deployed on hardware that supports only integer calculations. Providing similar examples of how the proposed approach is optimized for hardwares would strengthen the paper’s claims.

- Typos (minor):
    - In section 5.5, authors mentioned the proposed method is not suitable for more powerful "high-end systems, such as GPT", which I guess should be referring to "GPU".

**Questions:**

As mentioned in the weaknesses above, it would be helpful to clarify the following questions for the reviewers:
- How does the proposed combination of three techniques differ from or improve upon previous approaches that have used subsets of these methods? Providing more detail on this could help highlight the originality of the work.
- Intuition and Mathematical Justification of the Inverted Computational Complexity Ratio:
    - It would be useful to provide more discussion on the design of the K_model in the Inverted Computational Complexity Ratio. Specifically, the ratio is based on a weighted summation of j_i \times b_i. Is there an assumption that deeper layers contribute more to the efficiency improvement metric? Any intuition or mathematically justification is welcome.
    - The ratio shows a linear correlation with the number of operations. What is the definition of the number of operations in this context?
    - If the authors intend to reflect reduction in computational cost due to low-bit quantization, it would be helpful to display the number of "quantization bits" in table 5. Why is this proposed ratio a better metric than using the number of parameters, FLOPs, and quantization bits to measure the efficiency of the compressed model? Adding more discussion is helpful for readers to understand the advantage of this new metric compared to existing traditional metrics.
- In the result section, it could be helpful if the authors provide a more nuanced discussion of their model's strengths and weaknesses compared to these baselines, particularly in cases where P-BERT underperforms.

---

### Official Review · Reviewer_nXbq · 2024-10-27

**Soundness:** 1
**Presentation:** 2
**Contribution:** 1
**Rating:** 3
**Confidence:** 5

**Summary:**

This paper tackles the problem of optimizing Transformer-based models like BERT for hardware deployment. The authors propose a multi-level optimization using pruning, quantization, and knowledge distillation, aiming to make the BERT model more compact while preserving high accuracy on the target task. Furthermore, the paper introduced a novel metric, namely, Inverted computational complexity, to quantify the model’s computation requirements. The optimization process is conducted via a genetic algorithm to explore a predefined set of pruning and quantization parameters for the BERT model.

**Strengths:**

- The paper tackles an interesting problem for the ML research community—optimizing large models like BERT for hardware deployment on resource-constrained devices.
- The discussion of different optimization techniques (quantization, pruning, and KD) is quite engaging and well-explained.

**Weaknesses:**

- The paper needs more novelty. The proposed optimization for BERT directly applies previous and well-known techniques (i.e., quantization, pruning, and KD). The authors didn't well explain why leveraging the three techniques altogether and not focusing on exploring one technique (e.g., pruning). Overall, the paper seems more like a direct application of existing methods without further improvement. To enhance the paper's novelty, the authors may consider discussing in details the benefits from each optimization technique or exploring opportunities to develop a new algorithm that more effectively integrates different optimization techniques in a way tailored specifically to the BERT architecture.
- The paper also lacks proper discussion from an architectural point of view, specifically regarding the type of layers being pruned or quantized (e.g., attention or FFN). Different from CNNs, quantization or pruning in Transformers is not straightforward and needs careful consideration, especially at the attention layer level. However, the authors consider all layers the same and didn't explain why and how quantization/pruning is applied to the BERT layers. To strengthen the paper's contribution, the authors should include a breakdown of the impact of pruning and quantization on attention layers versus feed-forward layers and then explain their rationale for treating all layers uniformly.
- The authors posit strong claims on their proposed Inverted computational complexity metric without theoretical or empirical evidence. First, the metric is formulated as a product of the layer number, pruning rate, and number of bits, which are also the parameters being explored by the genetic algorithm. What type of information (if any) is being extracted from this product needs to be clarified. For example, what's the utility of the layer number'  i' in equation (6)? Second, the pruning rate and number of bits depend on the layer's type, which hasn't been discussed in subsection 4.3. Overall, to demonstrate the utility and credibility of the proposed metric, the authors must (i) theoretically discuss the metric computation in equation (6), (ii) conduct an ablation study (with different combinations of the metric's components), (iii) compare their metric against established hardware performance indicators (e.g., latency and memory) to show its practical relevance, and (iv) Discuss how the proposed metric accounts for different layer types, given that pruning and quantization may affect them differently.
- The discussion in 4.3.2 needs to be more convincing since the results shown in Figures 3 and 4 cannot be generalized because of the limited number of observations (scatter points). Additionally, while the authors claim their metric is better than FLOPs because quantization is not included in the latter, both metrics are not a good proxy for hardware efficiency estimation [1]. This is because efficiency is specific to the hardware architecture and type of operations. For the authors to justify their claim, an ablation study could be conducted to compare their proposed metric and a FLOPs-aware quantization (where each layer’s FLOPs is multiplied by the number of bits).
- There’s no comparison with the latest existing works on BERT model optimization with quantization [2], pruning [3], or knowledge distillation [4]. Without a comprehensive comparison with these SOTA works it’s hard to draw any tangible conclusion on the effectiveness and novelty of the proposed approach. Overall, the paper should discuss how P-BERT differs from or improves upon [2, 3, 4]. The authors could add a comparison table that includes their method alongside SOTA approaches, highlighting key differences and improvements.

**References:**
- [1]: Dehghani, Mostafa, et al. "The efficiency misnomer." arXiv preprint arXiv:2110.12894 (2021).
- [2]: Shen, Sheng, et al. "Q-bert: Hessian based ultra low precision quantization of bert." Proceedings of the AAAI Conference on Artificial Intelligence. Vol. 34. No. 05. 2020.
- [3]: Liu, Zejian, et al. "EBERT: Efficient BERT inference with dynamic structured pruning." Findings of the Association for Computational Linguistics: ACL-IJCNLP 2021. 2021.
- [4]: Muhamed, Aashiq, et al. "CTR-BERT: Cost-effective knowledge distillation for billion-parameter teacher models." NeurIPS Efficient Natural Language and Speech Processing Workshop. 2021.

**Questions:**

- Is there a specific rationale for combining all three techniques (quantization, pruning, and KD) in the proposed optimization, and would focusing on a single technique be more beneficial?
- How do the authors justify treating all layers within BERT similarly for quantization and pruning, given the distinct characteristics of attention layers and feed-forward networks (FFN)?
- What theoretical or empirical justifications are provided for the formulation of the Inverted Computational Complexity metric, especially concerning the inclusion of layer number as a factor?
- How does the dependence of pruning rate and number of bits on the layer type impact the reliability of the proposed metric, and why was this not addressed in subsection 4.3?
- Have the authors conducted an ablation study or provided theoretical evidence to demonstrate the credibility of their proposed metric and justify the specific choices made in equation (6)?
- Given the limited number of observations shown in Figures 3 and 4, how confident are the authors in the generalizability of their results, and could a larger dataset or different experimental setup change these findings?
- What comparisons can the authors provide to validate the superiority of their metric over FLOPs-based metrics in terms of hardware efficiency, especially considering the specificity of efficiency to hardware architecture and operation types?
- How does the proposed approach compare with the latest state-of-the-art works in BERT model optimization involving quantization, pruning, or knowledge distillation?

---

### Official Review · Reviewer_F4Ge · 2024-11-02

**Soundness:** 2
**Presentation:** 2
**Contribution:** 2
**Rating:** 3
**Confidence:** 4

**Summary:**

The paper explores the combination of weights pruning, bit quantization, and teacher-student learning via knowledge distillation in compressing BERT models. They used a genetic algorithm to determine the cluster of weights to be pruned and quantized. They proposed a new metric, the inverted compuational complexity ratio, which is defined roughly as the ratio of computation required bewteen a standard BERT and a compressed model, to capture the extent of compression. They evaluated the trade-off bewteen compression and accuracy on several Hugging Face classification tasks.

**Strengths:**

The use of a genetic algorithm to select compressed models based on the evaluation metric and the newly proposed inverted computation complexity ratio (ICCR) is seldom studied. It is an encouraging direction to explore.
The overall flow of the paper is easy to follow.
The comparison of the proposed P-BERT with other optimized BERT model on the selected evaluation tasks is well summarized.

**Weaknesses:**

The main compression techniques employed by the team are all common model compression techniques: pruning, quatization, and KD. Only a new genetic algorithm to search for prunable parameters do not seem sufficiently novel to me. The authors could conduct ablation studies to compare each of the three techniques in the P-BERT framework to help us understand what trade-offs can be made.
The ICCR is conveniently defined to describe the extent of compression, but it fails to capture how the compression is achieved (through pruning vs quantization, which layers, etc.). This could have hindereed the authors from interpreting the inconsistenties observed in Tables 1-4.
The presentation of the paper is another aspect that can be improved.
1. The references are cited in a rather unusual manner. Include the references in parentheses will help readers.
2. Tables and figures should not be bewteen texts (Fig. 1, 2, 3, 4, 7, 8; Table 5).
3. The authors can help readers understand the results better by interpreting the results by referencing to their figures and tables. For example, Section 5.2.1 only describes the outcome, but the readers are left to understand the significance of those plots. I personally did not get what the authors mean to convey here.
4. Some hyperparameters are given (without intution or prior knowledge, e.g. Section 5.2, 5.3). Some claims are laid without reference (Section 2, Section 4.3.2).
5. Appendix can be part of the paper.

**Questions:**

Why MSE is chosen over MAE in Eq. 3 and 4?
Why do we need to consider L_hard, L_soft, L_hidn, and L_attn in Eq. 5? What's the significance of each? Any ablation study to compare which ones are more important?
Figure 1, flow chart, why the sum of absolute? Any intuition or reference?
For the definition of K_model in Eq. 6, why do we need to include the layer index "i"? Does this mean that the last few layers carry more weights?
Why is the particular genetic algorithm chosen in Sect. 4.4?
What challenges did you run into in Sect. 5.4 when comparing against those models?
Would you like to consider palettization as an additional compression technique?

---

### Official Review · Reviewer_LMFk · 2024-11-03

**Soundness:** 2
**Presentation:** 2
**Contribution:** 2
**Rating:** 5
**Confidence:** 3

**Summary:**

The paper combines pruning, quantization and knowledge distillation to reduce the computational complexity of the BERT models. A genetic algorithm is used to hone down on the parts of the layers to prune and the appropriate precision of the layers for quantization. Knowledge distillation is later used to transfer knowledge from the baseline BERT model. They also use Inverted Computational Complexity Ratio (ICCR) as a metric to evaluate model compression factor. Results section shows the trend of decreasing estimated inference time with increasing ICCR for four different benchmarks. There are also comparisons of the performance of P-BERT’s estimated inference time and accuracy against other BERT models.

**Strengths:**

The proposed technique is a generalized methodology to explore pruning and quantization strategies for a given model and task. The Pareto front and the loss curve plots are quite insightful.

**Weaknesses:**

The results presented in the paper fail to emphasize the utility of this approach. Table 5 presents other models with higher complexity ratios and lower inference time than what’s best achieved by P-BERT. Even for CoLA where P-BERT achieves the best accuracy among tuned models, the compression factor doesn’t translate to reduction in inference time.

If the chosen experimental hardware setup isn’t able to leverage the pruning and quantization benefits, then a different evaluation platform or metric could be selected.

**Questions:**

Is the proposed methodology of using the genetic algorithm to prune and quantize applicable to other models?

Is the computational complexity metric in Eq 6 able to capture the significance of the deep layers? Could you provide some details like layerwise precision and % of parameters pruned of one of the resultant models?

---

### Official Review · Reviewer_wTeb · 2024-11-04

**Soundness:** 2
**Presentation:** 3
**Contribution:** 2
**Rating:** 5
**Confidence:** 3

**Summary:**

This paper presents P-BERT which is an optimization of the standard BERT model. It is developed through the integration of three model compression techniques: pruning, quantization, and knowledge distillation. It also introduces a novel metric, the Inverted Computational Complexity Ratio (ICCR), to better capture model efficiency and complexity. Experimental results demonstrate that P-BERT achieves a reduction in computational complexity of at least 60% while maintaining comparable accuracy to the baseline BERT across several natural language processing tasks.

**Strengths:**

+ It targets BERT optimization, which is important.
+ It proposes a novel metric ICCR, which provides a new way of evaluating the model efficiency.
+ The flow charts (Fig 1 and 5) and examples (Fig 2) can help understand the paper.
+ It provides the setting of hyper-parameters and is open-sourced.

**Weaknesses:**

- P-BERT performs well in CoLA, while its performances on other tasks are less competitive.
- The genetic algorithm is time-consuming and may get stuck in a local optimum. Therefore, giving enough reasons and motivations to choose the generic algorithm would be better.
- Knowledge distillation can guarantee the accuracy of the model but may be time-consuming.
- It would be better to provide more evaluations to show that the ICCR metric fits well for BERT optimization.

**Questions:**

1. In Fig 1, if a particular hidden state has already been marked for pruning in the previous layer, then in the current layer, the counter should simply increment by 1, and this hidden state should not be pruned again. However, in Fig 2, for layer 2, why is S_{768} still selected for pruning?
2. In Equation 6, why does it use the number of layer i? Why does it assign greater weights to the deeper layers?

---

### Note · Authors · 2025-01-20

I have read and agree with the venue's withdrawal policy on behalf of myself and my co-authors.